# Amber Extract Reduces Lipid Content in Mature 3T3-L1 Adipocytes by Activating the Lipolysis Pathway

**DOI:** 10.3390/molecules26154630

**Published:** 2021-07-30

**Authors:** Erica Sogo, Siqi Zhou, Haruna Haeiwa, Reiko Takeda, Kazuma Okazaki, Marie Sekita, Takuya Yamamoto, Mikio Yamano, Kazuichi Sakamoto

**Affiliations:** 1Faculty of Life and Environmental Sciences, University of Tsukuba, Tsukuba, Ibaraki 305-8572, Japan; erisuhcyn.nago@gmail.com (E.S.); zhousiqi66@gmail.com (S.Z.); t-yamamoto@kohaku-lab.com (T.Y.); 2Kohaku Bio Technology Co., Ltd., Tsukuba, Ibaraki 305-8572, Japan; h-haeiwa@kohaku-lab.com (H.H.); r-takeda@yamanobeautychemical.com (R.T.); k-okazaki@yamanobeautychemical.com (K.O.); m-sekita@yamanobeautychemical.com (M.S.); sakazbo@gmail.com (M.Y.)

**Keywords:** amber, adipocytes, lipid metabolism, lipolysis, hormone-sensitive lipase

## Abstract

Amber—the fossilized resin of trees—is rich in terpenoids and rosin acids. The physiological effects, such as antipyretic, sedative, and anti-inflammatory, were used in traditional medicine. This study aims to clarify the physiological effects of amber extract on lipid metabolism in mouse 3T3-L1 cells. Mature adipocytes are used to evaluate the effect of amber extract on lipolysis by measuring the triglyceride content, glucose uptake, glycerol release, and lipolysis-related gene expression. Our results show that the amount of triacylglycerol, which is stored in lipid droplets in mature adipocytes, decreases following 96 h of treatment with different concentrations of amber extract. Amber extract treatment also decreases glucose uptake and increases the release of glycerol from the cells. Moreover, amber extract increases the expression of lipolysis-related genes encoding perilipin and hormone-sensitive lipase (HSL) and promotes the activity of HSL (by increasing HSL phosphorylation). Amber extract treatment also regulates the expression of other adipocytokines in mature adipocytes, such as adiponectin and leptin. Overall, our results indicate that amber extract increases the expression of lipolysis-related genes to induce lipolysis in 3T3-L1 cells, highlighting its potential for treating various obesity-related diseases.

## 1. Introduction

Obesity is closely related to various diseases, such as type II diabetes, hypertension, cancer, cardiovascular diseases, and renal insufficiency [1]. Therefore, preventing or controlling obesity is essential to prevent the aforementioned related diseases.

Phytochemicals such as polyphenols and flavonoids are secondary metabolites necessary for plants to adapt to their environment [2]. Phytochemicals exert various physiological effects on our body, including antioxidant effects, which lead to the prevention of obesity [3]. Vegetables, fruits, and tea leaves, which have high phytochemical content, are known to reduce the risk of cardiovascular diseases [4,5]. Certain flavonoids have also been investigated that can lower the risk of breast cancer [6] and type II diabetes [7]. As mentioned above, natural products rich in phytochemicals are expected to reduce excessive obesity and contribute to the prevention and mitigation of various diseases.

Amber is the fossilized resin of trees, such as pine and cedar, often found on the Baltic coast. It mainly contains the monoterpenes succinic and rosin acids [8]. In addition to these compounds, amber contains numerous other components. Commonly, amber is used in jewelry or ornaments, but in Russia and Scandinavia, it has been used as an antipyretic and anti-inflammatory agent, as well as incense because of its unique sweet smell that is emanated when it is heated. Recently, amber has been shown the potential to prevent Alzheimer’s disease [9], and it has been proved to reduce the inflammation reaction in our lab [10]. Therefore, we hope to further study whether amber can regulate lipid metabolism inside adipocytes, and to investigate the mechanism.

Lipolysis, the process of lipid metabolism, is initiated by the secretion of catecholamines and activation of protein kinase A (PKA) caused by intense exercise or nutritional deficiencies. Activated PKA phosphorylates hormone-sensitive lipase (HSL), adipose triglyceride lipase (ATGL), and perilipin [11], which induce lipolysis. Activated ATGL degrades triacylglycerols (TAGs) present in lipid droplets into fatty acids (FAs) and diacylglycerols (DAGs) [12], which is followed by the action of HSLs that degrade DAGs into monoacylglycerols (MAGs) and FAs [13]. Subsequently, by the action of monoacylglycerol lipase, MAG is degraded into FA and glycerol [14], and glycerol is then released into the blood to be taken up by other organs.

In this study, amber extract was used to treat mature adipocytes, and the effect of amber on lipid metabolism was investigated. Furthermore, through mRNA and protein analyses, we preliminarily elucidated the mechanism through which lipid metabolism is regulated by amber.

## 2. Results

### 2.1. Amber Reduced the Accumulation of Lipid Droplets in Mature 3T3-L1 Cells

In this study, we focused on the effect of amber extract on lipid metabolism in mature adipocytes. Firstly, the toxicity of amber to mature 3T3-L1 cells was measured using the MTT assay. Mature adipocytes were treated with or without amber (concentration ranging from 10 to 100 µg/mL) for 96 h, and the concentration of ethanol in the culture medium was adjusted to 5%. At concentrations below 75 µg/mL (until 50 µg/mL), no cytotoxicity was observed, whereas 100 µg/mL of amber extract resulted in marked cytotoxicity (Figure 1A). Therefore, concentrations of 10, 25, and 50 µg/mL of amber extract were employed in further assays.

Next, the effect of amber extract on the total triglyceride levels in cells was determined by Oil Red O assay. Because the highest concentration of amber extract was reduced to 50 µg/mL, the concentration of amber in the extract was also reduced to 2.5%, which was the same in further experiments. Treatment of cells with all concentrations of amber extract reduced the number of triglycerides stored in the fat droplets (Figure 1B,C). These results indicated that amber extract treatment reduced the accumulation of lipid droplets in mature adipocytes without any cytotoxicity.

### 2.2. Amber Inhibited Glucose Uptake and Promoted Glycerol Release in Mature 3T3-L1 Cells

To determine the mechanism through which amber extract regulates fat accumulation in mature 3T3-L1 cells, the amount of glucose uptake and glycerol release up to 24 h after treatment with amber extract was assessed (Figure 2). The medium was collected after 6, 12, 18, and 24 h of amber treatment, and the amount of glucose and glycerol in the medium was measured. Because we measured the glucose content in the culture medium, the reduction in glucose content in the medium compared to that at 0 h was considered the amount of glucose uptake. After 18 and 24 h of treatment, glucose content in the medium treated with amber extract was higher than that in the untreated groups, thus implying that glucose uptake was significantly inhibited by treatment with 50 µg/mL of amber extract compared to that in the untreated control (Figure 2A). In contrast, glycerol content in the medium increased in a dose-dependent manner following amber extract treatment (Figure 2B). These results indicated that amber extract inhibited glucose uptake and promoted the release of glycerol.

### 2.3. Treatment with Amber Extract Altered the Expression of Factors Involved in Lipolysis

Next, we examined the changes in the expression of HSL, perilipin, and ATGL at the mRNA and protein levels following the treatment of cells with amber extract. ATGL is a lipase acting on TAG that triggers lipolysis followed by degradation by other lipases [12,13,14]. Treatment with amber extract tended to increase the expression of ATGL at the mRNA level (Figure 3A), with the highest expression observed at a concentration of 50 µg/mL of the amber extract. Perilipin enhances lipolysis by phosphorylating HSL in the lipid droplet membrane [15]. Treatment with amber extract altered the expression of perilipin at the mRNA level, which was significantly increased when the highest concentration of amber extract (50 µg/mL) was used (Figure 3B). The expression of perilipin at the protein level was also markedly increased amber extract treatment (Figure 3C).

HSL, along with ATGL and MGL, is a lipase that regulates lipolysis; phosphorylated HSL is transferred onto the lipid droplet membrane, where its enzymatic activity is enhanced [16]. The expression of HSL at the mRNA and protein levels was slightly increased (*p* > 0.05) by amber extract treatment (Figure 3D,E). In addition, the phosphorylation of HSL (Ser563, Ser565, and Ser660) was examined. The phosphorylation of Ser565 and Ser660 was increased (Figure 3G,H), whereas that of Ser563 was slightly decreased in the cells treated with amber extract (Figure 3F). These results indicated that amber extract promoted lipolysis by increasing the expression of perilipin-coding gene and inducing the phosphorylation of HSL.

### 2.4. Administration of Amber Altered the Secretion of Adipocytokines

Additionally, we investigated whether amber extract affects the expression of adipokines released by adipocytes. Adiponectin is known to prevent atherosclerosis [17], and diabetes [18]. Interestingly, the expression of adiponectin has a negative relationship with obesity [19]. The expression of adiponectin at the mRNA level showed an increasing trend. Still, a clear increase in its expression at the protein level was observed following treatment with different concentrations of amber extracts (Figure 4A,B). Leptin is a protein secreted by white adipocytes, and it regulates food intake, energy expenditure, and energy balance in rodents and humans [20]. Amber extract treatment decreased the expression of leptin at the mRNA level in a dose-dependent manner (Figure 4C). These results indicated that the amber extract could regulate the secretion of adipokines.

## 3. Discussion

In this study, the treatment of mouse 3T3-L1 cells with ethanolic amber extract induced lipolysis. In addition to ethanol, we used other solvents for extraction, such as methanol. In our comparative pre-study experiments on the ability of solvent-extracted amber to promote lipolysis, we found that ethanolic amber extract showed a better inhibitory effect than other extracts (data not shown). In addition, by considering the toxicity of the other extraction solvents, ethanol was chosen for extraction.

It is necessary to investigate the active components in amber extract, and our previous studies have attempted to identify the components using HPLC. Succinic acid, a major component of amber, has been reported to increase the production of heat by brown adipose tissue [21] and suppress lipid accumulation [22]. Recently, it has been reported that succinate accumulation in hypoxic adipose tissue contributes to lipolysis and insulin resistance [23], suggesting that exogenous succinic acid could promote lipolysis. However, further experiments are required to confirm the effects of succinic acid on lipolysis and clarify the mechanism. Moreover, since many other unknown components exist in amber, future studies should focus on identifying all components in amber extract and elucidating the exact component that promotes lipolysis.

In the present study, we focused on the regulatory effect of amber extract on lipid metabolism in differentiated adipocytes. Differentiation of 3T3-L1 cells is one of the most commonly used in vitro models to study adipose tissue biology [24]. Regrettably, in our pre-study experiments, we did not observe that ethanolic amber extract exerted a significant regulatory effect on the differentiation of 3T3-L1 cells. Therefore, in the present study, we only used mature adipocytes.

The results of the present study revealed that amber extract increased the expression of lipolysis-related genes encoding ATGL, HSL, and perilipin (Figure 3) and promoted lipolysis by inducing the phosphorylation of HSL and perilipin. ATGL is an essential lipase for the hydrolysis of TAGs [12]; HSL moves from the cytoplasm to the surface of lipid droplets when phosphorylated upon catecholamine stimulation [25], and lipid hydrolysis is promoted by phosphorylated perilipin upon stimulation with catecholamine [26]. We also detected the phosphorylation of HSL (Figure 3F–H). HSL has several serine residues that are phosphorylated by PKA. The phosphorylation of Ser660 induces the migration of HSL from the cytoplasm to the fat droplet membrane and increases its enzymatic activity [27]; Ser563, similar to Ser660, is phosphorylated during lipolysis, but does not directly affect HSL activity. During lipolysis, the phosphorylation of Ser660 is preceded by that of Ser563, but the role of Ser563 phosphorylation is not yet completely understood [11]. In addition, Ser565 is not phosphorylated during the induction of lipolysis, but is rather phosphorylated during the normal state (when lipolysis does not occur) and is thought to play a role in preventing lipolysis [28]. We observed that with respect to the phosphorylation of HSL, amber extract treatment increased the phosphorylation of Ser660 and Ser565, but decreased the phosphorylation of Ser563 (Figure 3F–H). The original mechanism underlying lipolysis would have promoted lipolysis by increasing the phosphorylation of Ser660 and Ser563 and decreasing the phosphorylation of Ser565, but this was not the case in the present study. We observed that amber extract treatment increased the phosphorylation of Ser660 and Ser565 by approximately 1.5–1.8- and 1.2–1.3-fold, respectively. In contrast, the phosphorylation of Ser563 was not markedly altered upon treatment with amber extract treatment, and it decreased by 0.8–0.9 fold. These results suggested that treatment with amber extract increases the phosphorylation of Ser660, which increases the migration of HSL to the fat droplet membrane and enhances its enzymatic activity. The timing of dephosphorylation and phosphorylation of Ser565 and Ser563 is not yet clear, and therefore, the upstream signaling activity that causes HSL phosphorylation should be investigated in the future.

Fatty acid is also released together with the lipolysis and glycerol release [12,13,14]. We also measured the non-esterified fatty acid (NEFA) in the medium, and the increased NEFA content inside the medium was investigated after 12 h and 24 h treatment (Appendix A). However, we found the NEFA content was time-dependently decreased in the control group (Appendix A). Therefore, it is difficult to explain the function of amber in promoting the lipolysis or inhibiting the NEFA uptake, which is necessary to investigate the reason in the future.

As an important endocrine organ, adipose tissue regulates systemic metabolic homeostasis and responds to nutrient flux to meet the metabolic demands of positive or negative energy balance. Numerous secretory factors derived from adipocytes, such as adiponectin and leptin, play a role in maintaining metabolic homeostasis [29]. To analyze the effect of amber extract treatment on adipokine secretion, we tested the expression of genes encoding adipokines at 96 h of treatment. Previous studies have reported that patients with obesity-related diseases have low adiponectin levels [19]. Moreover, administering exogenous adiponectin or inducing its overexpression in cells can reduce the symptoms of the disease [30]. The results of the present study showed that the expression of adiponectin increased significantly after 96 h of amber extract treatment, which indicated that amber could be used as a natural alternative to insulin-sensitizing or anti-atherogenic agents.

Leptin, one of the peptides secreted by white adipose tissue, has been known to control appetite and body weight and regulate energy balance [20]. The secretion of leptin is regulated by the fat mass and adipocyte size. Therefore, the level of leptin is elevated in obese patients [31]. Some in vivo experiments have shown that leptin promotes lipolysis; however, the function of leptin is mainly accomplished by activating the sympathetic nervous system, rather than directly acting on adipose tissue [32]. A previous study also reported phenomena similar to those demonstrated in the current study, i.e., promotion of lipolysis (Figure 2B) and suppression of leptin expression (Figure 4C) [33]. Therefore, our results indicate that amber regulates total fat mass by promoting lipolysis, thereby reducing the expression of leptin.

Plasminogen activator inhibitor (PAI-1) is a known adipokine that facilitates thrombus production and promotes atherosclerosis, and its expression increases with the progression of obesity [34]. In the previous study, recombinant adiponectin directly inhibited PAI-1 expression in 3T3-L1 adipocytes, indicating that an increase in adiponectin expression results in a decrease in PAI-1 levels [35]. Here, amber extract treatment increased the expression of PAI-1 at the mRNA level, while it weakly decreased its secretion into the medium (data not shown). The secretion of PAI-1 and expression of adiponectin (Figure 4A) are consistent with the conclusions of previous studies. However, since the components in amber have not yet been fully investigated, we cannot deny the existence of a component that can promote the expression of PAI-1, which will be identified in future studies.

In the current study, we found that amber extract not only promoted lipolysis, but also inhibited the uptake of glucose. Because glucose metabolism and insulin function are closely related [36], we are presently conducting in vivo studies using mice. We also intend to test the regulatory effects of amber extract treatment on adipose tissue generation and lipid metabolism in mice; in addition, we seek to examine the effects of amber extract treatment on glucose metabolism and insulin function in mice fed a high-fat diet.

## 4. Materials and Methods

### 4.1. Amber Extract

Fifty grams of amber from Kaliningrad, Russia, was crushed and powdered and then added to 400 mL of 50% ethanol. It was extracted at 40 °C for 1 h with stirring and then filtered. The residue was extracted again with 50% ethanol, as described above, and filtered. The filtrate was subjected to depressurizing and freeze-drying to obtain it in a powdered form (Kohaku Bio Technology, Tsukuba, Japan). This powder was dissolved in 50% ethanol to a concentration of 1000 µg/mL, and the solution was used as the mother stock of amber extract.

### 4.2. Cell Culture

The mouse progenitor adipocyte line 3T3-L1 (Health Science Research Resources Bank, Osaka, Japan) was used for the experiments. 3T3-L1 cells were cultured in a normal culture medium (i.e., Dulbecco’s Modified Eagle Medium (DMEM) high-glucose (Sigma-Aldrich, Burlington, MA, USA)) supplemented with 10% fetal bovine serum (Hyclone, Logan, UT, USA) and incubated at 37 °C and 5% CO_2_.

The full differentiation medium consisted of a normal culture medium and DMI. The components of DMI were as follows: 10 µg/mL insulin (FUJIFILM Wako Pure Chemical Corporation, Osaka, Japan), 1 µM dexamethasone (Sigma-Aldrich, Burlington, MA, USA), and 500 µM 3-isobutyl-1-methylxanthine (Sigma-Aldrich, Burlington, MA, USA). The full differentiation medium was used to induce the differentiation of 3T3-L1 cells for the first 2 days. In the next 2 days, a normal culture medium containing 5 µg/mL insulin was used to induce the differentiation process. In the last 4 days, a normal culture medium was refreshed, and mature adipocytes were obtained at day 8 after DMI stimulation. After the maturation of cells, a normal culture medium was used for further experiments.

### 4.3. Cytotoxicity Test

Mature adipocytes were treated with amber extract for 96 h. After 96 h, MTT assay was performed by adding 10% thiazolyl blue tetrazolium bromide (MTT) solution ((5 mg/mL in PBS) MTT in DMEM high-glucose), and the cells were then cultured for 4 h at 37 °C. After 4 h, the medium was removed, and isopropanol containing 0.04 M HCl was added to dissolve formazan. Then, the absorbance at 570 nm was measured using the microplate reader Epoch2 (BioTek Instruments Japan, Tokyo, Japan). Cell viability was determined relative to the absorbance of the control group in which the cells were not treated with the amber extract.

### 4.4. Oil Red O Assay

Mature 3T3-L1 cells treated with amber extract for 96 h were fixed with paraformaldehyde (Wako Pure Chemical Industries, Ltd., Osaka, Japan) at 4 °C overnight. Then, the medium was removed, and the cells were washed with PBS twice. Oil Red O solution (3 mg/mL Oil Red O (Sigma-Aldrich, Burlington, MA, USA) in a 3:2 mixture of isopropanol and double-distilled water) was added, and the cells were cultured at room temperature for 10 min. After 10 min, the medium was removed, and cells were washed with PBS four times. Then, isopropanol was added, and the absorbance was measured at 490 nm.

### 4.5. Glucose Uptake

The effect of amber extract on the amount of glucose taken up by mature 3T3-L1 cells was quantified using the LabAssayTM glucose kit (Wako Pure Chemical Industries, Ltd., Osaka, Japan). The amount of glucose remaining in the medium was measured. The amount of glucose in the medium compared with that present in the fresh medium was recognized as the amount of glucose uptake.

### 4.6. Glycerol Release

The effect of amber extract on the amount of glycerol released from mature 3T3-L1 cells into the medium was quantified using the LabAssayTM Triglycerides Kit (Wako Pure Chemical Industries, Ltd., Osaka, Japan). The amount of glycerol released into the medium was calculated by measuring the absorbance of the medium at 600 nm.

### 4.7. Real-Time Polymerase Chain Reaction (RT-PCR)

RNA from 3T3-L1 cells treated with amber extract was extracted using RNA iso Plus (Takara Bio, Shiga, Japan). cDNA was synthesized using the PrimeScriptTM RT reagent kit with gDNA Eraser (Perfect Real Time, Takara Bio Inc., Kusatsu, Shiga, Japan), followed by the addition of THUNDERBIRD^®^ SYBR^®^ qPCR Mix (Toyobo, Osaka, Japan). PCR was performed using a Thermal Cycler Dice^®^ Real Time System Lite (Takara Bio Inc., Kusatsu, Shiga, Japan). The tubes were set up in an RT-PCR machine, and the reaction was performed under the following conditions: 95 °C for 30 s, followed by 40 cycles at 95 °C for 5 s and 60 °C for 30 s, and then at 95 °C for 15 s, 60 °C for 30 s, and 95 °C for 15 s. GAPDH was used as a standard internal gene. Amplification of target mRNAs was performed using specific primers (5′–3′):

ATGL (sense: TGCTGGAGGCCTGTGTGGAA; antisense: TCAGGGACATCAGG CAGCCACT); HSL (sense: AAGACCACATCGCCCACAGC; antisense: GCTGTCTGAAG GCTCTGAGTTGC); perilipin (sense: CTCTGGGAAGCATCGAGAAG; antisense: GCATGGTGTGTCGAGAAAGA); adiponectin (sense: GTGGATCTGACGACACCAAAAG G; antisense: AACGTCATCTTCGGCATGACTGG); leptin (sense: TATGTTCAAGCACTGCCT; antisense: CTTGAGTCAAGCATTGTC); GAPDH (sense: TGGTGAAGGTCGGTGTGAACGG; antisense: TGCCGTTGAATTTGCCG TGAGT)

### 4.8. Western Blot Analysis

3T3-L1 cells were seeded onto a 6-cm dish and differentiated to adipocytes. Cells were lysed using RIPA buffer (50 mM Tris-HCl (pH 7.4), 150 mM NaCl, 1% Triton X-100, 0.1% SDS, 0.5% Na-deoxycholate, 1 mM EDTA, 10 mM NaF, 1 mM Na_3_VO_4_, 0.8 µM aprotinin, 50 µM bestatin, 20 µM leupeptin, 10 µM pepstatin A, and 1 mM AEBSF (Wako Pure Chemical Industries, Ltd., Osaka, Japan)). Then, cells were disrupted using an ultrasonic homogenizer and centrifuged to collect the supernatant. The proteins in the cell lysate were quantified using a BCA protein assay kit (Wako Pure Chemical Industries, Ltd., Osaka, Japan). Next, 20 µg of protein from each sample was electrophoresed and transferred to a PVDF membrane (ATTO, Tokyo, Japan). The membrane was incubated with primary and secondary antibodies (Cell Signaling Technology, Inc., MI, USA) and developed using the LumicoGLO reagent (Cell Signaling Technology, Inc., MI, USA). Then, the bands were detected and photographed using an AE-9300H Ez-Capture MG (ATTO, Tokyo, Japan).

### 4.9. Statistical Analysis

The statistical significance of all the data were determined using the Student’s *t*-test and are denoted as * *p* < 0.05, ** *p* < 0.01, and *** *p* < 0.001.

## 5. Conclusions

The ethanolic extract of amber reduced lipid accumulation in matured 3T3-L1 adipocytes, resulting in the induction of lipolysis. Moreover, amber extract treatment increased the expression of perilipin and HSL and their phosphorylation. Treatment with amber extract also increased the expression of adipocytokines. Overall, the present study indicates that amber extract increases the expression of lipolysis-related genes to induce lipolysis in 3T3-L1 cells, highlighting its potential for treating various obesity-related diseases.

## Figures and Tables

**Figure 1 molecules-26-04630-f001:**
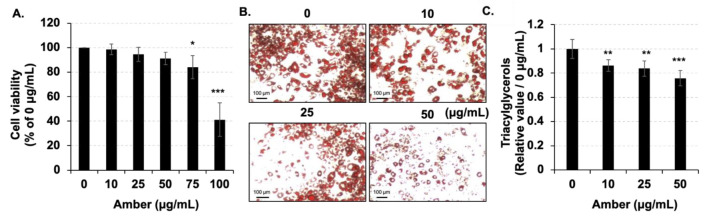
Effects of amber extract on cell viability and lipid accumulation. (**A**) After treatment of 3T3-L1 cells with amber extract (0, 10, 25, and 50 µg/mL) for 96 h, cell viability was measured using the MTT assay. (**B**) Photographs of lipid droplets in cells stained with Oil Red O stain. (**C**) Triacylglycerol (TAG) levels in the lipid droplets were measured using Oil Red O staining. The results are presented as the mean ± SD; *n* = 3; * *p* < 0.05, ** *p* < 0.01, *** *p* < 0.001 vs. 0 µg/mL.

**Figure 2 molecules-26-04630-f002:**
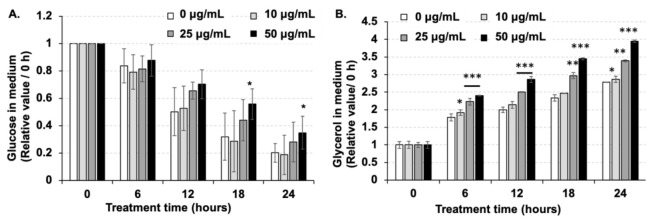
Effects of amber extract on lipid metabolism. After mature adipocytes were treated with amber extract (0, 10, 25, and 50 µg/mL) for 6, 12, 18, and 24 h, the culture medium was collected. The amount of (**A**) glucose and (**B**) glycerol in the medium at 6, 12, 18, and 24 h after treatment with amber extract was measured. The decrease in glucose content in the medium compared with that at 0 h was recognized as the amount of glucose uptake. The results are indicated as the mean ± SD; *n* = 4; * *p* < 0.05, ** *p* < 0.01, *** *p* < 0.001 vs. 0 µg/mL.

**Figure 3 molecules-26-04630-f003:**
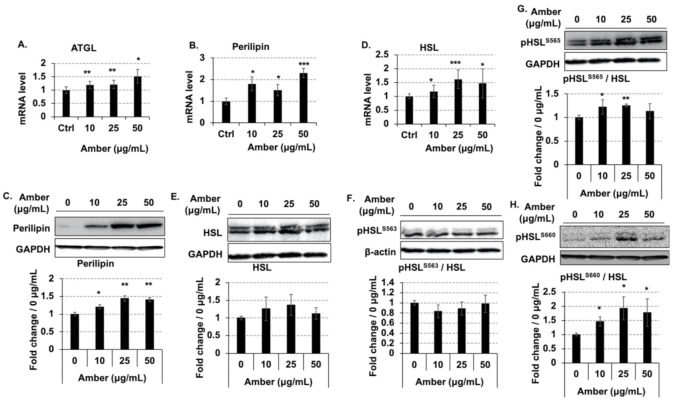
Effects of amber extract on lipolysis-related gene expression. Mature 3T3-L1 cells treated with amber extract (0, 10, 25, and 50 µg/mL) were collected after 24 h, in which mRNA levels were measured by RT-PCR, and protein levels were detected by western blotting. Expression of (**A**) adipose triglyceride lipase (ATGL), (**B**) perilipin, and (**D**) hormone-sensitive lipase (HSL) at the mRNA level was detected by RT-PCR; the data were directly obtained from the machine. Western blot images of (**C**) perilipin, (**E**) HSL, (**F**) pHSL^S563^, (**G**) pHSL^S565^, and (**H**) pHSL^S660^ analyzed by ImageJ. Statistical analysis of western blot bar graphs was performed in three independent experiments. The results are expressed as the mean ± SD; *n* = 3; * *p* < 0.05, ** *p* < 0.01, *** *p* < 0.001 vs. 0 µg/mL.

**Figure 4 molecules-26-04630-f004:**
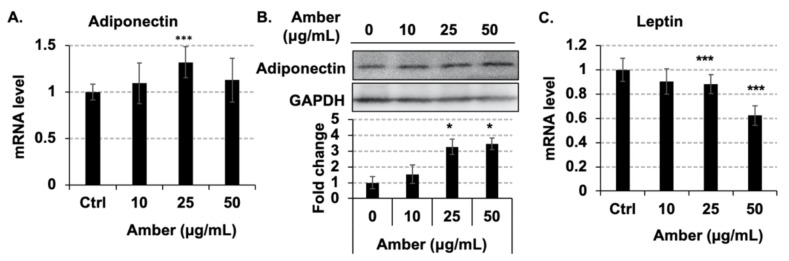
Effects of amber extract treatment on adipocytokine-associated gene expression in 3T3-L1 cells. Mature 3T3-L1 cells treated with amber extract (0, 10, 25, and 50 µg/mL) were collected after 96 h, in which mRNA levels were measured by RT-PCR, and protein levels were detected by western blotting. Expression of (**A**) adiponectin and (**C**) leptin at the mRNA level was detected by RT-PCR; the data were directly obtained from the machine. Western blot images of (**B**) adiponectin analyzed by ImageJ. Statistical analysis of western blot bar graphs was performed in three independent experiments. The results are presented as the mean ± SD; *n* = 3; * *p* < 0.05, *** *p* < 0.001 vs. 0 µg/mL.

## Data Availability

Data is contained within the article or Appendix A.

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
