# Peer review of "Amber Extract Reduces Lipid Content in Mature 3T3-L1 Adipocytes by Activating the Lipolysis Pathway"

_molecules, 2021, doi:10.3390/molecules26154630_

Round 1
Reviewer 1 Report
This manuscript is acceptable for publication.
Author Response
Thank you for reviewing our paper.

Reviewer 2 Report
In this paper “Amber Extract Reduces Lipid Content in Mature 3T3-L1 Adipocytes by Activating the Lipolysis Pathway”, the authors aimed to clarify the physiological effects of amber extract on lipid metabolism in mouse 3T3-L1 cells. Mature adipocytes were used to evaluate the effect of amber extract on lipolysis. The amber extract increases the expression of lipolysis-related genes to induce lipolysis in 3T3-L1 cells, highlighting its potential for treating various obesity-related diseases. I recommend this work to be published after major revision in the Molecules Journal. However, I have few comments and I recommend the authors address them. My comments are below.
- Clarify sentences 9-11 in the abstract. We cannot simply expect resins to have some physiological effects. Author can omit “it is expected” from the sentence and make it clearer based on some evidence.
- Include references for lines 44 and 45.
- Change the level in Figure 1C for triacylglycerols on y-axis.
- Did authors measure free fatty acids in the medium? Please provide the data of FFA content in the medium for few samples such as cells treated with 0 and 50 µg/mL. Increased glycerol in the medium could suggest increased FFA is the medium. Authors could use any assay to measure it.
Author Response
Please find the responses in the attachment.

Round 2
Reviewer 2 Report
Authors have responded well to my comments.
Thanks
This manuscript is a resubmission of an earlier submission. The following is a list of the peer review reports and author responses from that submission.
Round 1
Reviewer 1 Report
The authors tried to demonstrate the effects of amber extract on lipid metabolism in 3T3-L1 cells.
Although the amber extract reduced lipid droplet and glucose uptake, and promoted glycerol release, it did not change the expression of most lipolysis-associated factors. Only the level of perilipin was significantly increased upon Amber treatment.
This manuscript needs further data (evaluation of fat browning, fatty acid oxidation, thermogenesis, etc) to support the effects of amber extract on lipolysis.
Results 2.3, and discussion : It is misinterpreting to state that amber treatment increased the expression of HSL, its phosphorylation, and adipocytokines, since the increase was not statistically significant.
Figure 4 legend : Check out the text.
Reviewer 2 Report
The article titled “Amber Extract Reduces the Lipid Contents in Mature 3T3-L1 Adipocytes by Promoting Lipolysis Pathway “ by Erica Sogo et al analyze the effects of amber extract on lipid metabolism in mouse 3T3-L1 cells, and measure the mRNA level of relevant genes.
This issue is interesting, however there are some concerns:
The authors present the effects of amber extract on lipid metabolism without the characterization of alcoholic extract. The authors should explain why they chose an alcoholic extract of amber and what is the chemical composition, quality and quantity of extract components.
The authors use an alcoholic extract at different concentrations and refer it to zero concentration. To exclude that the biological effect is due to amber components and not to ethanol, they should compare the effect of amber ethanol extract to the same concentration of ethanol alone.
Minor concernrs:
Fig. 1A: Significance is missing. Check if amber at 75 µg / mL is toxic, as the viability has been reduced to about 70%;
Fig. 3C: The authors should evaluate the expression at times less than 24 hours, considering that gene expression precedes the biological effect;
Fig. 3: The authors could explain why they test different treatment times between glucose release and gene expression;
Fig. 4: There is no caption.
Reviewer 3 Report
The following needs to be revised:
lane 74: ..(B) should be (C)
Figure 3C: is the treatment at 50ug/ml significant?
lane 109-111: the sentence is repeated twice.
Figure4. legend is missing.
Review figure for significance. Fo example, figure 4.